# AI Respondents for Policy Monitoring: From Data Extraction to AI-Driven Survey Responses in the OECD STIP Compass

## Abstract

Science, Technology, and Innovation (STI) policies are central to national and international competitiveness, yet their complexity makes systematic mapping and continuous monitoring a persistent challenge. This study draws on one of the largest initiatives in the field, the OECD STIP Compass survey, which collects and organizes data on STI policy from OECD countries and has historically relied on extensive manual survey efforts to ensure global consistency. Large Language Models (LLMs) are redefining representation learning in NLP, enabling them to process and internalize knowledge from long unstructured documents. This paper presents a novel application of LLMs for structured information extraction and generation from STI policy documents, focusing on OECD data across six sample countries. We develop a data extraction pipeline based on long-context in-context learning to encode task-specific schemas that allow learning of survey taxonomy labels from public URLs referencing policy initiatives. The pipeline integrates validation steps using a secondary LLM for relevance and evidence scoring, and comparison with survey responses completed by human respondents. For evaluation, we apply multiple overlap measures, including overlap ratios, agreement scores between human-generated and LLM-generated policy indicators, and K-fold cross-validation for AI-generated labels. Our findings indicate that LLMs can achieve high overlap with human respondents for policy indicators (84-95%). Qualitative analysis reveals that the model tends to provide more detailed descriptions, complementing human-written content. Our approach points to the potential of an AI-assisted framework for STI policy monitoring, enhancing both efficiency and quality in international policy intelligence.

## 1 Introduction

Science, Technology, and Innovation (STI) policies are complex socio-technical constructs that play a central role in shaping national competitiveness and addressing global challenges. Yet, systematic mapping and continuous monitoring of these policies remain costly and labor-intensive, particularly in the context of large-scale international surveys such as the EC-OECD STIP Compass (Flanagan et al., 2011). The Compass aggregates data on STI policy initiatives across OECD and partner countries, relying on expert respondents to fill in structured survey instruments linked to web-based sources of evidence. While this approach provides a unique comparative perspective, it faces challenges of scale, consistency, and timeliness as the number and complexity of initiatives expand.

Large Language Models (LLMs) are redefining natural language processing (NLP) by enabling machines to internalize knowledge from large unstructured corpora and to adapt to diverse downstream tasks through prompting rather than parameter updates (Tan et al., 2024; Mao et al., 2025; Dherin et al., 2025). Recent generative LLMs such as GPT-4o are capable of long-context reasoning and in-context learning, making them suitable for information extraction from extended policy documents and web content. Their ability to generate structured responses aligned with human-designed schemas offers a potential solution to the persistent difficulties of innovation policy data collection and validation.

However, integrating LLMs into international policy monitoring is not straightforward. The prior work shows that LLMs can act as "artificial respondents", replicating social science experiments by generating survey answers conditioned on demographic profiles (Ashokkumar et al., 2024). On the other hand, LLM-driven data generation may introduce systematic biases and distortions—so-called model collapse—if models are repeatedly trained on synthetic outputs (Shumailov et al., 2024). Moreover, innovation policy data pose unique challenges as policies are heterogeneous, multi-scalar, and embedded in institutional contexts that are not always captured in publicly available sources (Feldman et al., 2015).

This paper contributes to the emerging field of AI-assisted policy intelligence by presenting a novel application of LLMs for the EC-OECD STIP Compass. Specifically, we design and test a data extraction pipeline that uses long-context prompting to map survey taxonomy codes (policy instruments, themes, and target groups) from web-scraped content provided by survey respondents. A secondary validation layer employs an LLM to evaluate outputs on dimensions of relevance and evidence. Using a pilot across six OECD countries (Canada, Finland, Germany, Korea, Spain, and Türkiye), we assess the overlap between LLM-generated and human-generated survey responses and explore complementarities in descriptive and objective fields (Hajikhani et al., 2024). Our findings show that LLMs achieve high overlap in structured codes (84–95%) but diverge in textual fields, where AI tends to provide more detailed procedural descriptions while humans emphasize contextual and societal impacts. These insights highlight the promise of hybrid human-AI approaches for international policy monitoring. The key contributions of this paper are as follows:

- We design a novel data extraction pipeline that leverages long-context in-context learning (ICL) to process lengthy unstructured policy documents.

- We implement a validation layer that evaluates relevance and evidence by employing another LLM as a validator model.

- We evaluate the pipeline in a pilot study covering six OECD countries, analyzing overlap, agreement, and cross-validation between LLM-generated and human-generated responses.

## 2 RELATED WORK

The methodological challenges of collecting and comparing STI policy data have long been recognized in the literature on policy mixes and innovation systems. Policies are difficult units of analysis, and large-scale cross-country data are costly to compile and validate (Flanagan et al., 2011). Efforts to address these challenges have included international surveys and expert-driven databases such as the OECD STIP Compass, yet these approaches are constrained by reporting burden, data gaps, and inconsistencies across national contexts.

The rise of LLMs introduces new opportunities to address these challenges. LLMs have been applied successfully in tasks such as information extraction, summarization, and question answering, often outperforming earlier supervised NLP methods (Tan et al., 2024; Mao et al., 2025). Their in-context learning capabilities allow them to adapt dynamically to survey-style questions without the need for costly labeled training data (Dherin et al., 2025). Recent studies demonstrate the ability of LLMs to act as proxies for human subjects in social science experiments, suggesting their potential as scalable substitutes or complements to traditional survey respondents (Ashokkumar et al., 2024).

At the same time, concerns remain about their reliability. Shumailov et al. (2024) warn of distributional drift and degradation in model outputs when systems recursively train on synthetic data. In the context of STI policy, the absence of gold-standard labeled datasets and the heterogeneous nature of policy initiatives make fine-tuning approaches less feasible, as highlighted in recent experimentation with the STIP Compass (Hajikhani et al., 2024). Instead, long-context prompting combined with expert-designed taxonomies offers a pragmatic way to leverage LLMs while maintaining human oversight.

Our work builds directly on these strands by testing an operational pipeline that integrates LLMs into the STIP Compass survey process. While prior research has explored web-based policy document analysis and retrieval-augmented methods, the contribution here is to demonstrate, for the first time, a systematic comparison between human-provided survey responses and AI-generated responses across multiple dimensions of STI policy data. In doing so, we extend earlier calls to leverage the

"new data frontier" in innovation studies (Feldman et al., 2015) with AI-driven approaches that are both scalable and adaptable to international policy monitoring.

# 3 METHODOLOGY

In this section, we describe the data extraction pipeline for the EC-OECD STIP Compass survey. The raw data were obtained from the OECD and consists of the content from URLs that survey participants identified as relevant policy initiatives. The following subsections describe the preparation of the data for pre-filling, prompt design, and evaluation. Figure 1 illustrates the workflow of our methodology.

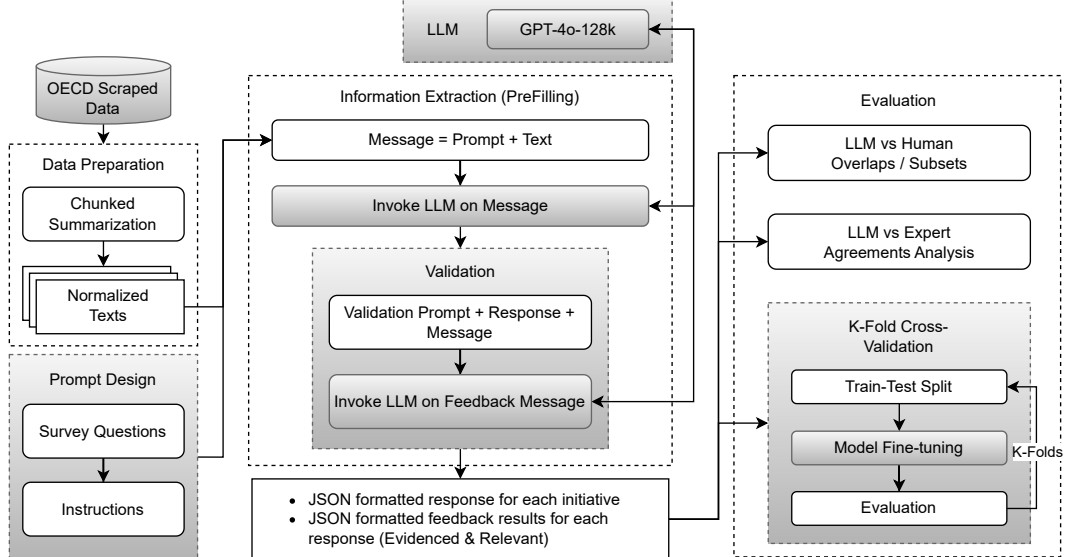

Figure 1: Methodological workflow of the study: starting from OECD-scraped policy texts, followed by chunked summarization and data preparation, prompt design with survey questions, and information extraction using GPT-4o-128k (OpenAI, 2024). Validation and evaluation involve comparing LLM-generated outputs with human annotations, as well as cross-validation using model fine-tuning.

## 3.1 DATA PREPARATION

We processed the survey data and the scraped text provided by the OECD to retain only those initiatives with sufficient content for analysis. Initiatives containing fewer than 200 tokens (less than 100 words) were discarded as insufficient for analysis. In contrast, initiatives with more than 120,000 tokens were further processed to fit within the LLM context window. For this purpose, we devised a chunked summarization method that divided the initiative content into chunks of 50,000 tokens and prompted an LLM to summarize each chunk while retaining underlying information related to STI policies. The resulting summaries were then aggregated by the LLM to produce a complete text containing all relevant initiative information. The prompts designed for the chunked summarization method are given below.

> *Individual Chunk Prompt: "The following is a set of texts related to a policy initiative: "+ chunk +" Based on this list of docs, write a detailed account covering description, objectives, dates, policy themes and instruments, key stakeholders, budgets, and evaluations. Capture exact details, examples, and cases for these dimensions. Be thorough, detailed, and comprehensive. Use only text from the provided documents."*

> *Chunk Summary Aggregation Prompt: "The following is a set of detailed summaries: "+ chunked_summaries +" Take these and generate a detailed account covering description, objectives, dates, policy themes and instruments, key stakeholders, budgets, and evaluations. Capture exact details, examples, and cases for these dimensions. Be thorough, detailed, and comprehensive. Use only text from the provided summaries."*

## 3.2 Pre-filling - Long-context In-Context Learning

Long-context In-Context Learning (ICL) refers to the ability of LLMs with extended context windows to generalize examples integrated in lengthy prompts, often spanning thousands of tokens (Bertsch et al., 2025). Retrieval-Augmented Generation (RAG), in contrast, integrates an LLM with an external retrieval system, typically querying a document index by employing a neural retriever, and prompts the model's generation on the retrieved content (Lewis et al., 2020; Asai et al., 2023). Through qualitative analysis, we found that RAG performs suboptimal for survey Pre-filling from long STI policy documents because; 1) the knowledge to be extracted from scraped policy content was not explicitly defined, making it difficult to formulate direct fact-based queries; and 2) adapting RAG by splitting survey questions into smaller prompts leads to redundancy, as overlapping policy indicators and guidelines produce overlapping content.

Long-context ICL enables the inclusion of complete content, survey questions, and extraction guidelines in a single extended prompt, leveraging the long context windows in the latest LLMs to preserve coherence and yield detailed responses. We adopted a long-context ICL that incorporated examples within the prompts to shape and improve the quality of responses. This approach captures a wide yet relevant range of material without restricting nuanced findings, making it well-suited for our study. The capacity of current large context window LLMs adequately accommodated most of the cases in our dataset.

### 3.2.1 Prompt Design

We incorporated survey questions and OECD guidelines into the prompt design. The primary objective of the prompt design was to ensure responses in English while accommodating multilingual cases. The guidelines covered all indicators, including descriptions and objectives, policy instruments (PI), target groups (TG) policy themes (TH), start date, budget, and evaluation report. We experimented with various arrangements of instructions, classifications, and examples in the prompts to ensure consistent and unified responses. Additionally, we designed the prompts to produce responses in a structured format for easier parsing and integration.

We adhered to a schema in which the LLM provided the context and predefined classifications, particularly for policy instruments, target groups, and policy themes. We ensured that the information identified from the raw scraped text was appropriately categorized. The prompt instructions emphasized avoiding the generation of new content and instead relying on the provided content. The list of designed prompts is provided in Appendix B.3.

### 3.2.2 Response Validation

Evaluation metrics have evolved with the growing popularity of LLMs (Gu et al., 2024; Li et al., 2024). A notable development is the use of LLM-based evaluation, where one model employs Chain-of-Thought (CoT) reasoning to assess the outputs of another LLM (Saha et al., 2025). This approach emphasizes dimensions such as groundedness and completeness through well-designed prompting strategies, which can then be scored numerically (Kim et al., 2023). Research indicates that larger model sizes generally produce improved performance in summarization evaluation, with stronger correlation to human judgments (Liu et al., 2023; Fabbri et al., 2021). In addition, evaluation methods may employ reference-based approaches that compare the generated text with the ground truth or reference texts (Wu et al., 2023).

To validate the generated responses, we employed another instance of the LLM to evaluate them against the prompt and source material. A binary (0/1) scoring scheme was applied to assess two factors: *evidence* and *relevance*. The model evaluates whether each response is evidenced by the text and whether the response is relevant to the instructions given. We developed a structured evaluation prompt (B.1) to ensure a consistent and objective assessment of the generated responses, focusing on their adherence to the source material and relevance to the instructions. This evaluation filtered out cases that could have resulted from hallucinations or misinterpretations.

### 3.3 Evaluations

In addition to response validation, we incorporated three further evaluation measures.

1. *Overlap analysis* provided a comparison between human- and LLM-generated survey responses, capturing the extent of alignment between the two datasets.

2. The naïve overlap calculation does not necessarily provide a meaningful insight into agreement, as it only reflects surface-level similarity. Therefore, *label-wise agreement* scoring is applied to quantify the consistency between the human annotations and model outputs with respect to policy labels using high-, medium-, and low-agreement categories.

3. Manual extraction of the structured policy indicators from long, unstructured documents is labor-intensive and prone to error. Moreover, the LLM-generated survey responses could not be fully validated due to the absence of gold-standard reference. To address this limitation, we validated LLM-responses against structured policy labels using *k-fold cross-validation* by fine-tuning masked and causal models.

## 4 EXPERIMENTAL SETUP

In our experiments, we performed survey pre-filling using LLMs as survey respondents to extract and generate multiple types of data fields. Our study addressed two key questions: (1) Can web-scraped content provide sufficient and relevant information to pre-fill STIP Compass survey questions? (2) Is it feasible to map unstructured web-scraped content to structure survey categories employing LLMs to generate survey responses in place of human respondents? We hypothesized that long-context LLMs could capture a significant portion of structured information for free-text fields as well as multi-label policy identification.

### 4.1 CHOICE OF LLM

We selected GPT-4o-128k (OpenAI, 2024) for our study because of its extended context window and strong performance across evaluation benchmarks. The selection was supported by the latest metrics from Stanford's Holistic Evaluation of Language Models (HELM), which provides a comprehensive assessment of language models' capabilities and limitations. According to the HELM leaderboard[1], GPT-4o (2024-05-13) achieved a mean win rate of 0.938 on standard evaluation metrics.

### 4.2 EVALUATION DESIGN

To validate the generated responses, we employed another instance of GPT-4o-128k to evaluate the responses against the prompt and raw material. We used a structured evaluation prompt (B.1) to ensure consistency in assessing the generated responses, focusing on their adherence to the source material and relevance to the instructions.

For post-extraction evaluations, we again employed GPT-4o-128k as an evaluator for free-text fields, including descriptions and objectives. We designed a prompt (B.2) to compare overlaps and discrepancies between human participants and LLM-generated responses. The prompt instructs the LLM to quantify the results into four categories: full overlap, high overlap, low overlap, and no overlap. However, the degree of overlap against policy labels was quantified using overlap percentages. We computed agreement scores using micro F1 scores throughout the dataset. Similarly, micro F1 scores were used to evaluate k-fold (k=5) cross-validation experiments by fine-tuning (system prompt B.4) a range of masked and causal models (Table 4).

### 4.3 IMPLEMENTATION DETAILS

Data preparation, pre-filling, response validation, and free-text field evaluation were conducted using Azure AI Services[2] by deploying different instances of GPT-4o. Running the experiments with GPT-4o incurred a total cost of €446.84. Dataset analysis and agreement scores were computed using standard Python libraries. In k-fold cross-validation, the dataset was shuffled and split into 80% training, 10% validation, and 10% testing for each fold. The main hyperparameters for masked and causal models, as well as LoRA configurations, are summarized in Appendix A.

---

[1]https://crfm.stanford.edu/helm/lite/latest
[2]https://ai.azure.com

## 5 RESULTS & DISCUSSION

The results of the study provided a systematic comparison between human-generative and LLM-generated survey responses in six sample OECD countries, focusing on free-text fields and policy related indicator labels. The following subsections present Human-LLM overlaps and subsets, agreement analysis of indicator labels, and k-fold cross-validation of LLM-generated labels.

### 5.1 DATASET ANALYSIS

Table 1 presents the country-level statistics after data preparation and filtering. The *Insufficient* column refers to the share of policy initiatives without URLs or containing less than 200 tokens. The *Unidentified* column presents initiatives that have sufficient content, but relevant policy instruments could not be extracted. The *Sufficient* column shows percentages of initiatives that have sufficient and suitable web content. The last column reports the number of samples with the appropriate STI content from policy initiatives.

Table 1: Distribution of web content and number of samples by country.

| Sr# | Country | Insufficient | Unidentified | Sufficient | # of Samples |
|-----|---------|--------------|--------------|------------|--------------|
| 1 | Canada | 30% | 6% | 64% | 149 |
| 2 | Finland | 32% | 11% | 57% | 80 |
| 3 | Germany | 25% | 7% | 68% | 193 |
| 4 | Korea | 27% | 20% | 53% | 146 |
| 5 | Spain | 31% | 13% | 56% | 142 |
| 6 | Türkiye | 47% | 12% | 41% | 135 |
| | Total | – | – | – | 845 |

Each sample contains eight policy indicators covering various types of content. *Policy instruments (PI)*, *Target groups (TG)*, and *Policy themes (TH)* contain underlying indicators in the form of sub-labels that refer to documented policies. Table 2 presents statistics of indicator coverage comparing human-generated and LLM-generated labels.

Table 2: Comparison of label statistics between expert-generated and LLM-generated labels.

| Generated | PI Labels | Unique PI | TG Labels | Unique TG | TH Labels | Unique TH |
|-----------|-----------|-----------|-----------|-----------|-----------|-----------|
| By humans | 1,281 | 27 | 3,828 | 33 | 1,895 | 51 |
| By LLM | 2,336 | 28 | 4,727 | 33 | 3,013 | 57 |

Figure 2 shows the frequency distribution of policy labels by class. Both datasets exhibited similar label distributions, with a significant imbalance across classes. Many labels are underrepresented, with rare appearances in the dataset.

### 5.2 HUMAN-LLM OVERLAP

We investigated qualitative differences in free-text responses, identifying complementary tendencies in which LLMs delivered more detailed procedural accounts, while human respondents emphasized contextual and societal dimensions. Figure3 shows the overlap results for both free-text fields.

The analysis of overlap between descriptions indicated that the predominant share of cases (74.05%) exhibited high overlap, whereas only 1.19% demonstrated full overlap, 15.24% were classified as low overlap, and 9.52% showed no overlap. These differences suggest that human and AI assessments can complement each other by offering diverse perspectives and insights on the same topics. However, the objectives showed high overlap in 41% of the cases, while no overlap was observed in about 36% of the cases. Low and full overlap are less frequent in 22% and 1% of the cases, respectively. These patterns suggest that differences often stem from variations in approach, level of detail, scope, and available information for assessments.

We further examined the extent of overlap in multi-label indicators–policy instruments (PI), target groups (TG), and policy themes (TH)–to evaluate the representational accuracy of LLMs relative to human experts. Table 3 shows the overlap between labels provided by human participants and those

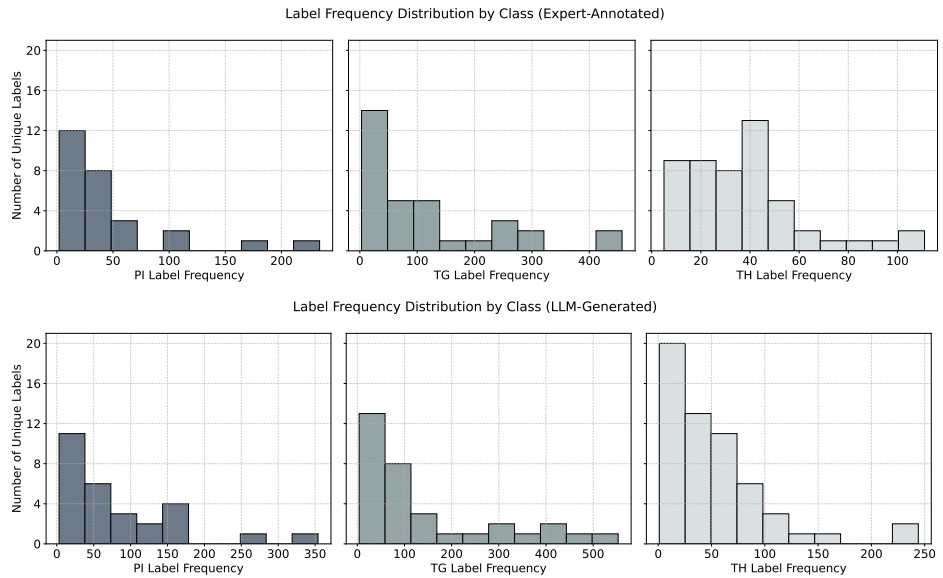

Figure 2: Comparison of policy indicator labels between expert-generated and LLM-generated.

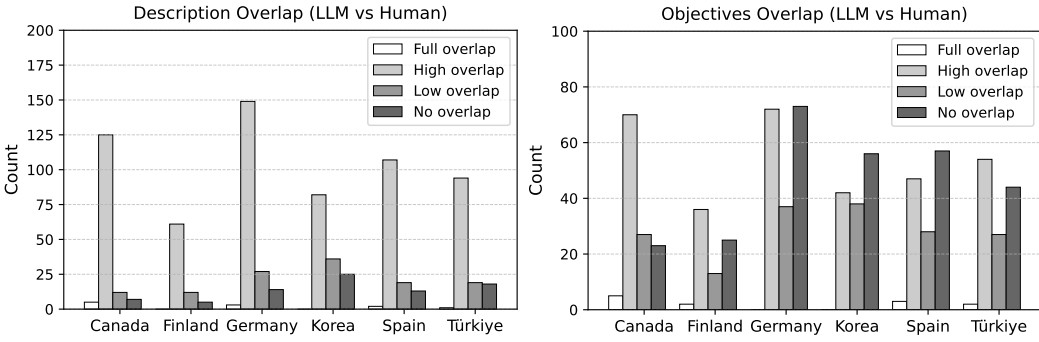

Figure 3: Overlap of survey participant responses and LLM responses per country on the policy initiative description and objectives.

generated by LLM. The table includes all overlapping cases where there is an overlap for at least one policy label. The LLM performed relatively well in capturing survey respondent codes for policy themes and instruments, and particularly well for policy target groups. On average, in 95% of policy initiatives the LLM identified at least one of the target groups provided by survey respondents. The corresponding averages are 84% for policy themes and 85% for policy instruments. Although some variation exists across countries, these differences are generally limited.

## 5.3 HUMAN-LLM AGREEMENT

Figure 4 presents the distribution of the label-wise agreement scores. The agreement scores shown in the graph are quite dispersed, ranging from high to low, reflecting the overall average level of agreement between human respondents and the LLM. Given that policy indicators span a wide range of dimensions, achieving consistently high agreement is challenging and depends on interpretation, comprehension, and background knowledge.

Our analysis indicates that policy indicators with clear and unambiguous definitions tend to yield higher level of agreement. The top three labels, one from each category, provide clear interpretation and are as follows: PI015: *"Indirect financial support|Tax or social contributions relief for firms investing in R&D and innovation"*, TG21: *"Net zero transitions|Net zero transitions in energy"*, PT92: *"Research and education organizations|Public research institutes"*. In contrast, low agree-

Table 3: Overlap of survey participant responses and LLM responses per country for policy instruments (A), target groups (B), and policy themes (C).

| Sr# | Country | Policy instruments (A) | Target groups (B) | Policy themes (C) |
|---|---|---|---|---|
| 1 | Canada | 84% | 97% | 85% |
| 2 | Finland | 84% | 98% | 84% |
| 3 | Germany | 80% | 97% | 83% |
| 4 | Korea | 88% | 93% | 84% |
| 5 | Spain | 85% | 94% | 82% |
| 6 | Türkiye | 88% | 93% | 87% |
| | Total | 85% | 95% | 84% |

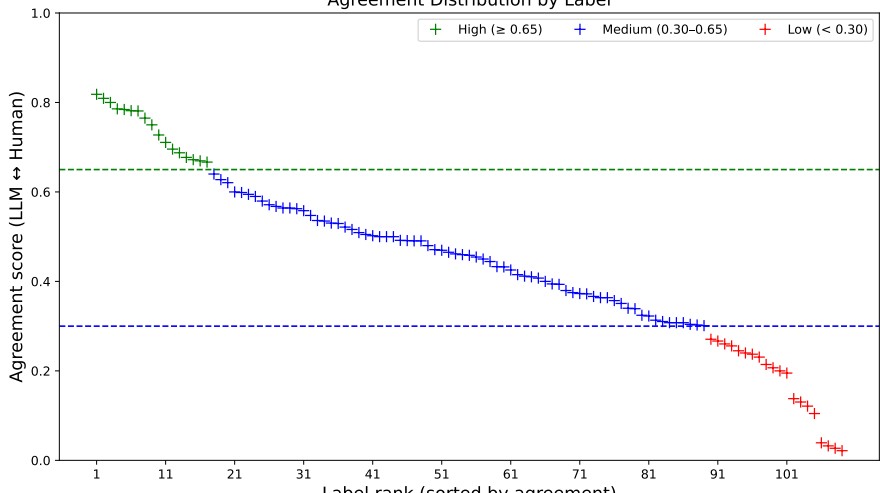

Figure 4: Human vs LLM agreement score distribution. High agreement scores are shown in green, medium scores in blue, and low scores in red.

ments are mainly due to abstract, broad, overlap in categories, and general terminologies in policy definitions. Three labels, one from each category with lowest score, are as follows: PI010: *"Direct financial support|Procurement programmes for R&D and innovation"*, TG25: *"Firms by age|Firms of any age"*, TH16: *"Public research system|Public research debates"*. These patterns highlight that the clarity and specificity of policy indicator definitions play a decisive role in shaping the degree of agreement between human respondents and the LLM.

## 5.4 CROSS-VALIDATION

The cross-validation results are reported using micro F-measures, as shown in Table 4. The experiments were conducted with several open-source and closed-source LLMs.

Table 4: F-Fold cross-validation micro F1 scores for LLM-generated policy indicators.

| Model | Precision | Recall | F1 |
|---|---|---|---|
| RoBERTa-large (Liu et al., 2019) | 86.70 | 66.61 | 75.33 |
| BigBird-RoBERTa-base (Zaheer et al., 2020) | 87.04 | 69.25 | 77.12 |
| BigBird-RoBERTa-large (Zaheer et al., 2020) | 88.43 | 76.01 | 81.74 |
| Llama-3.1-8B-Instruct (Grattafiori et al., 2024) | 82.02 | 90.79 | 86.18 |
| Mistral-7B-Instruct-v0.3 (Mistral-AI, 2023) | 92.91 | 90.90 | 91.89 |
| GPT-OSS-20B (OpenAI, 2025) | 97.63 | 97.83 | 97.73 |
| GPT-3.5-Turbo 16k (OpenAI, 2023) | 98.33 | 97.62 | 97.98 |
| GPT-4o 128k (OpenAI, 2024) | 98.14 | 98.04 | 98.09 |

RoBERTa achieved only average F-scores, primarily due to its limited input length, whereas its BigBird variants performed better due to their extended input length and block-sparse attention mechanism. In contrast, causal models demonstrated a stronger ability to capture information from longer documents and achieved higher F-scores in multi-class classification. These results highlight the importance of model architecture and context length in determining performance on complex policy classification tasks.

## 5.5 DISCUSSION

The findings of this study demonstrate that large language models (LLMs) can act as effective "artificial respondents" for the OECD STIP Compass, offering both efficiency and depth in survey data collection. The comparison between human and LLM-generated responses highlights strong complementarities rather than simple substitution. Specifically, while LLMs tend to provide more detailed procedural and descriptive accounts, human experts emphasize the contextual and societal implications of policy initiatives.

The overlap analysis shows that LLMs achieve high accuracy in structured indicators, with agreement levels of 84–95% across policy instruments, target groups, and themes. However, in free-text fields such as initiative descriptions and objectives, divergences remain. Only 1.19% of the cases reached full overlap, while the majority (74.05%) demonstrated high but not identical overlap. This suggests that LLMs capture the formal aspects of initiatives reliably but may not fully replicate the nuanced framing and interpretive perspectives provided by human respondents.

These patterns highlight the potential of hybrid approaches: LLMs can reduce reporting burdens by pre-filling surveys with structured and detailed information, while human experts can refine, contextualize, and interpret these responses. Nevertheless, risks remain. Over-reliance on synthetic LLM outputs may lead to biases, redundancy, or "model collapse" if such outputs are recursively integrated into training data . Ensuring continuous human oversight and triangulation with original sources is thus essential for the long-term integrity of international policy monitoring.

Overall, the evidence supports the viability of integrating AI into the STIP Compass workflow. Doing so would not only improve scalability and reduce costs but also enhance the descriptive richness of policy monitoring—provided safeguards are in place to preserve contextual accuracy and mitigate systemic biases.

## 6 CONCLUSION

This study introduced a novel LLM-based pipeline for policy monitoring within the OECD STIP Compass, demonstrating that AI can substantially complement human expertise in large-scale international surveys. By leveraging long-context in-context learning and a secondary validation layer, the approach achieved high overlap with human-generated responses across structured indicators, while providing additional procedural detail in free-text fields. The results highlight three key findings:

1. **Efficiency gains** – LLMs can significantly reduce manual reporting burdens by reliably pre-filling structured survey categories.
2. **Complementary perspectives** – LLMs enrich the descriptive layer of policy initiatives, while human respondents provide necessary contextualization and societal framing.
3. **Scalability with safeguards** – Hybrid human-AI systems can improve international policy intelligence, but careful oversight is required to address risks of bias, redundancy, and over-reliance on synthetic outputs.

Future work should expand the scope beyond the six pilot countries, refine validation mechanisms, and explore how human-AI collaboration can be systematically embedded into STI policy monitoring. Ultimately, the integration of LLMs into the STIP Compass marks a step toward more scalable, consistent, and timely global policy intelligence, paving the way for evidence-based innovation governance at the international level.

**Use of LLMs:** We acknowledge the use of ChatGPT-5 for writing assistance, grammar polishing, and improving clarity.

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

## A HYPERPARAMETERS

Table 5: Hyperparameters for masked (encoder) models

| Setting | Value |
|---------|-------|
| Max sequence length | 512 / 1600 |
| Batch size (train/eval) | 8 / 8 |
| Learning rate | 3e-5 |
| Epochs | 40 |
| Folds | 5 |
| Weight decay | 0.01 |
| Precision | FP16 |
| Eval/save strategy | Per epoch; best model (f1_micro) |
| Threshold (sigmoid) | 0.5 |
| Label filter | Frequency $\geq 5$ |

Table 6: Hyperparameters for causal models

| Setting | Value |
|---------|-------|
| Max input length | 7500 |
| Max new tokens | 200 |
| Batch size (train/eval) | 2 / 2 |
| Gradient accumulation | 8 (effective batch $\approx 16$) |
| Learning rate | 2e-5 |
| Epochs | 4 |
| Folds | 4 |
| Scheduler | Cosine; warmup ratio 0.03 |
| Precision | bfloat16 |
| Data collator | Causal LM (no MLM) |
| Label filter | Frequency $\geq 5$ |

Table 7: LoRA adapter hyperparameters (causal models)

| Parameter | Value |
|-----------|-------|
| $r$ (rank) | 8 |
| lora_alpha | 32 |
| lora_dropout | 0.05 |
| bias | none |
| task_type | CAUSAL_LM |
| Target modules | q_proj, k_proj, v_proj, o_proj |

## B PROMPTS

### B.1 VALIDATION PROMPT

```
Evaluate the Response against the given Instructions
and Text.  Provide a 0/1 assessment for the following
dimensions:  'evidenced' and 'relevant'.  Use the
following criteria to assess 'evidenced':  Is the
Response evidenced in the Text?
0 - No, there is no evidence supporting the Response
in the Text.
1 - Yes, there is evidence supporting the Response in
the Text.
Use the following criteria to assess 'relevant':  Is
the Response relevant to the Instructions?
0 - No, the Response is not relevant to the
Instructions (the Response does not follow or answer
the Instructions).
```

```
1 – Yes, the Response is relevant to the Instructions
(the Response does follow or answer the Instructions).
Structure your evaluation in a JSON format with two
keys: 'evidenced' and 'relevant'. 'evidenced' should
be the 0/1 assessment for whether the response is
evidenced in the text, and 'relevant' should be the
0/1 assessment for whether the response follows the
instructions. Do not elaborate or provide any further
explanation.
Example JSON structure:
{
'evidenced': 1,
'relevant': 1
}
Instructions: ``+ message +'' Response: + material
```

## B.2 FREE-TEXT EVALUATION PROMPT

```
"You are given two sets of policy initiative
documents: one is [human assessment] and the other
is [LLM assessment]. Your task is to analyze these
documents to understand their similarities, overlaps,
and differences using quantifiable metrics. Determine
the level of overlap and choose only one category from
"Full overlap", "High overlap", "Low overlap", or "No
overlap". Provide the response in the following JSON
format with an appropriate category and observation.
Example 1:
{
"Full overlap": {
"Observation": "Both assessments agreed entirely on
the need for increased funding for education."}
}
Example 2:
{
"High overlap": {
"Observation": "Both assessments focus on promoting
RDI activities, increasing competitiveness, and
attracting foreign investments. However, the LLM
assessment provides a more detailed breakdown of these
objectives."}
}
Example 3:
{
"Low overlap": {
"Observation": "The LLM emphasized renewable energy
incentives more than the human assessment."}
}
Example 4:
{
"No overlap": {
"Observation": "The human assessment discussed
healthcare reforms, which were not mentioned by the
LLM."}
}
```

### B.3 PRE-FILLING PROMPTS

*Identification*
Using the information provided, determine whether
[REPLACE] is discussed. Respond 1 if yes and 0 if
no. If you cannot determine whether the text contains
information about [REPLACE], respond 99 and do not
elaborate. Only respond with 0, 1, or 99.

*Description*
Act as an expert policy analyst. Based on the
given policy-related text material provide a short
description in English of [REPLACE] in sentence
format, not exceeding 100 words. Avoid using jargon;
be concise and clear, delivering only information
retrieved from the text. If there is no discussion
or mention of the topic, respond with "No information"
and do not elaborate. Provide the response in a JSON
format where the suggested name or theme is the key
and the description is the value.
Example JSON structure:
{
"description": "Description of the relevant policy
content in a clear and concise sentence."
}

*Objectives*
Act as a policy expert identify the [REPLACE]
initiative's objectives discussed. Structure your
response in JSON format with two keys: 'objective'
and 'description'. 'objective' should be a short
title of the objective and 'description' should be
a brief explanation of the objective, not exceeding
100 words. Both should be provided in English. If
you did not find any objective just indicate "n.a."
Example JSON structure:
{
"objective": "Enhance International Profile",
"description": "To give public research institutes
a higher profile in the international context by
providing funding for international collaboration."
}

*Start date*
Using the information provided, which is a
policy-related document, your task is to determine
the starting date of the [REPLACE] initiative if
mentioned. Structure your answers as a JSON file
where the key is the date which can be year and month
and the value is the description of what the date
refers to in English. Avoid using jargon; be concise
and clear, delivering only information retrieved from
the text. If there is no discussion or mention of the
topic, respond with "n.a." and do not elaborate.
Example JSON structure:
{
"2024-12": "Start date of the new environmental
regulation initiative.",
"2023-06": "Launch date of the public health

756  awareness campaign.",
757  "n.a.":  "No starting date mentioned for the
758  initiative."
759  }
760
761  *Policy instruments*
762  Consider yourself an expert policy analyst tasked with
763  reading documents related to Science Technology and
764  Innovation (STI) policy.  Your goal is to comprehend
765  and identify which of the below instrument(s) the
766  [REPLACE] initiative is relevant to and then assign
767  them to relevant policy instrument types based on
768  the information within the given policy instrument
769  labels and policy instrument definitions.  These
770  categories are provided to you in a JSON file with
771  keys such as "policyInstrumentID" for the ID of that
772  policy instrument, "label" for the descriptive label
773  of the policy instrument, and "definition" for the
774  description of what the policy instrument is and
775  what it relates to.  Based on the text given to
776  you, identify which definitions of the given policy
777  instrument types fit and are mentioned in the text,
778  and identify the Policy Instrument ID with your short
779  justification in English.  A text can be relevant to
780  one or more Policy Instrument ID, return only relevant
781  matches.  You should structure your response as a JSON
782  array where each object contains "PolicyInstrumentID"
783  as the key for the policy instrument ID and "reason"
784  as the key for your reasoning.  If you don't find any
785  relevant Policy Instrument, just say "n.a."
786  Example response format:
787
788
789  [
790    {
791      "PolicyInstrumentID": "PI019",
792      "reason": "reasoning..."
793    },
794    {
795      "PolicyInstrumentID": "PI020",
796      "reason": "reasoning..."
797    }
798  ]
799
800  If no relevant Policy Instrument is found, the
801  response should be:
802  {
803  "n.a.":  "No relevant Policy Instrument found."
804  }
805
806  Here are the Policy Instrument Types, labels, and
807  their definitions in a structured JSON file:
808  ```json
809  {
     "policyInstrumentTypes":  [
     {
     "PolicyInstrumentID":  "PI024",
     "label":  "Governance|Strategies, agendas and plans",
     "definition":  "Strategies that articulate the
     government's vision regarding the contribution of

```
Science technology and innovation to social and
economic development.  They set priorities for public
investment in STI and identify the focus of government
reforms, for instance in areas such as funding of
public research and promoting business innovation."
},
{
"PolicyInstrumentID":  "PI030",
"label":  "Governance|Creation or reform of governance
structure or public body",
"definition":  "Significant changes in the
institutional arrangements concerning STI policy
processes.  Possible examples include mergers of
STI-related ministries, reform of an innovation agency
or creation of a new oversight body."
},
{
"PolicyInstrumentID":  "PI031",
"label":  "Governance|Policy intelligence (e.g.
evaluations, benchmarking and forecasts)",
"definition":  "Tools for advancing policy learning
that aim to improve the design and implementation of
policies or that seek to fine-tune STI governance
arrangements.  Possible examples include policy
evaluations, benchmarking studies, system reviews,
technology assessments and foresight exercises."
},
{
"PolicyInstrumentID":  "PI025",
"label":  "Governance|Formal consultation of
stakeholders or experts",
"definition":  "Programmes allowing non-government
actors (e.g.  the research community, business, civil
society, regional and local governments) to express
their views or provide expert advice that inform
policy-making processes."
},
{
"PolicyInstrumentID":  "PI026",
"label":  "Governance|Horizontal STI coordination
bodies",
"definition":  "Public body ensuring the coherence
of STI policy making by setting up mechanisms to
co-ordinate different levels of governments.  For
instance, research and innovation councils and
committees may mediate between different ministries
and agencies, provide policy advice, set policy
priorities and/or oversee policy evaluation."
},
{
"PolicyInstrumentID":  "PI033",
"label":  "Governance|Regulatory oversight and ethical
advice bodies",
"definition":  "Dedicated authorities or publicly
funded boards that assess, monitor and/or advise on
the implementation or need for formal regulations soft
law or ethical frameworks accounting for technological
developments.  Examples include data protection
authorities and bioethics committees."
```

```
864          },
865          {
866          "PolicyInstrumentID":  "PI027",
867          "label":  "Governance|Standards and certification for
868          technology development and adoption",
869          "definition":  "Support provided for the development
870          and adoption of local and international standards,
871          including metrology, inspection, certification,
872          accreditation and conformity assessments."
873          },
874          {
875          "PolicyInstrumentID":  "PI028",
876          "label":  "Governance|Public awareness campaigns and
877          other outreach activities",
878          "definition":  "Instruments promoting the awareness
879          of STI activities and entrepreneurial and innovation
880          culture within non-governmental actors.  Examples
881          include science fairs in public schools and open days
882          in universities or power plants."
883          },
884          {
885          "PolicyInstrumentID":  "PI006",
886          "label":  "Direct financial support|Institutional
887          funding for public research",
888          "definition":  "Non-competitive grants funding HEIs
889          and PRIs according to various criteria (e.g.  research
890          capacity and performance indicators) to fulfil their
891          research missions.  Block funding provides these
892          organisations with stable resources and a certain
893          degree of autonomy in their research activities."
894          },
895          {
896          "PolicyInstrumentID":  "PI007",
897          "label":  "Direct financial support|Project grants for
898          public research",
899          "definition":  "A direct allocation of funding to HEIs
900          or PRIs seeking to finance all or part of a research
901          project.  Grant schemes can vary from very simplistic,
902          one-off funding allocations, to complex strategic
903          programs built on formal public-private partnerships."
904          },
905          {
906          "PolicyInstrumentID":  "PI008",
907          "label":  "Direct financial support|Grants for
908          business R&D and innovation",
909          "definition":  "A direct allocation of funding to
910          firms seeking to finance all or part of a project
911          involving R&D and/or innovation activities.  Grant
912          schemes can vary from very simplistic, one-off funding
913          allocations, to complex strategic programs built on
914          formal public-private partnerships."
915          },
916          {
917          "PolicyInstrumentID":  "PI009",
             "label":  "Direct financial support|Centres of
             excellence grants",
             "definition":  "Competitive grants funding the
             core activities of higher education and public
             research institutes and focusing on the promotion
```

```
of high quality scientific research.  Funding may be
associated to a performance contract."
},
{
"PolicyInstrumentID":  "PI010",
"label":  "Direct financial support|Procurement
programmes for R&D and innovation",
"definition":  "The process whereby public bodies
commission R&D activities or innovative goods and
services from third parties.  These bodies may
include government agencies at different national
and sub-national levels, as well as state-owned
enterprises."
},
{
"PolicyInstrumentID":  "PI011",
"label":  "Direct financial support|Fellowships and
postgraduate loans and scholarships",
"definition":  "Initiatives providing financial
support to encourage researchers to establish careers
in public sector research and industry (fellowships)
and for higher education students at master's level or
above (loans and scholarships)."
},
{
"PolicyInstrumentID":  "PI012",
"label":  "Direct financial support|Loans and credits
for innovation in firms",
"definition":  "Government-subsidised programmes that
allow firms to raise working or investment capital
by borrowing under better conditions compared to
the market.  Subsidised loans and credits are often
geared toward specific objectives, such as export
promotion (i.e.  export credit) or the acquisition
of new equipment."
},
{
"PolicyInstrumentID":  "PI013",
"label":  "Direct financial support|Equity financing",
"definition":  "Government-subsidised investment in
which small and innovation-intensive companies sell
equity (shares) to raise capital.  They use this
capital to fund their growth, as they often have
limited capacity to generate revenue at this early
stage of the entrepreneurial process."
},
{
"PolicyInstrumentID":  "PI014",
"label":  "Direct financial support|Innovation
vouchers",
"definition":  "Vouchers are small grants allocated
to SMEs to purchase services from external knowledge
providers.  Vouchers are often employed to fund
business advisory and technology extension services,
among others."
},
{
"PolicyInstrumentID":  "PI015",
"label":  "Indirect financial support|Tax or social
```

```
contributions relief for firms investing in R&D and
innovation",
"definition":  "Incentives that reduce the tax burden
of firms who invest in eligible R&D and innovation
activities, representing an indirect way of financial
support.  Examples include corporate tax income
benefits, reductions in tariffs for imported research
equipment, reimbursements of value added tax and
reductions to social insurance contributions."
},
{
"PolicyInstrumentID":  "PI016",
"label":  "Indirect financial support|Tax relief for
individuals supporting R&D and innovation",
"definition":  "Incentives that reduce the tax burden
of individuals who donate monies to public research
activities (e.g.  conducted by universities) or who
directly invest in R&D and innovation activities (e.g.
R&D intensive start-up)."
},
{
"PolicyInstrumentID":  "PI029",
"label":  "Indirect financial support|Debt guarantees
and risk sharing schemes",
"definition":  "Schemes working to cover some portion
of the losses experienced by lenders when firms
default on loans.  These are widely used as financial
instruments for supporting SME growth."
},
{
"PolicyInstrumentID":  "PI021",
"label":  "Collaborative infrastructures (soft and
physical)|Networking and collaborative platforms",
"definition":  "Instruments aiming to gather together
actors within the innovation system.  For instance,
entrepreneurs, investors and companies sharing common
geographical locations.  Another example includes
science-industry platforms seeking to support the
commercialisation of knowledge."
},
{
"PolicyInstrumentID":  "PI022",
"label":  "Collaborative infrastructures (soft and
physical)|Dedicated support to research and technical
infrastructures",
"definition":  "Instruments that support the creation
of new facilities, resources and services used by
the science community and Research and Technology
Organisations (RTOs) to conduct research and
foster innovation.  They include major scientific
facilities, demonstration and testing facilities,
e-infrastructures such as data and computing systems
and communication networks."
},
{
"PolicyInstrumentID":  "PI023",
"label":  "Collaborative infrastructures (soft
and physical)|Information services and access to
datasets",
```

```
        "definition":  "Online platforms providing access
        to collections of data on research and innovation
        activities.  This includes resources such as archives
        or scientific data and directories of actors in a
        given innovation ecosystem."
        },
        {
        "PolicyInstrumentID":  "PI017",
        "label":  "Guidance, regulation and
        incentives|Technology extension and business advisory
        services",
        "definition":  "Instruments that support innovation
        and entrepreneurship activities by stimulating
        improvements in businesses.  These may cover aspects
        such as operations, production, quality, logistics,
        workforce skills, learning capabilities and the
        adoption of new technologies and often have the
        objective of increasing firm productivity and
        efficiency."
        },
        {
        "PolicyInstrumentID":  "PI032",
        "label":  "Guidance, regulation and incentives|Science
        and technology regulation and soft law",
        "definition":  "Laws, rules, guidelines, directives
        or other policies made by a public authority on
        the development or use of new technologies (e.g.
        artificial intelligence, biotechnology, quantum
        computing) or practices in science.  Examples include
        the General Data Protection Regulation (GDPR) and
        bioethics legislation and scientific codes of
        conduct."
        },
        {
        "PolicyInstrumentID":  "PI018",
        "label":  "Guidance, regulation and incentives|Labour
        mobility regulation and incentives

        ",
        "definition":  "Instruments that promote the
        recruitment across sectors and/or countries of
        highly qualified individuals including scientists and
        engineers.  Sample initiatives include funding for
        international research projects, talent attraction
        programmes and coherent and efficient migration
        regimes."
        },
        {
        "PolicyInstrumentID":  "PI019",
        "label":  "Guidance, regulation and
        incentives|Intellectual property regulation and
        incentives",
        "definition":  "Instruments regulating and promoting
        the adoption of intellectual property rights and
        practices.  This includes the registration and
        commercialisation of intangible assets that are the
        result of human innovation and creativity."
        },
        {
```

```
        "PolicyInstrumentID":  "PI020",
        "label":  "Guidance, regulation and incentives|Science
        and innovation challenges, prizes and awards",
        "definition":  "A monetary (or other) incentive
        offered to STI actors in recognition of their
        contributions to research and innovation.  Inducement
        prizes reward a solution to a research/innovation
        challenge.  Recognition awards are ex-post prizes
        given to highly innovative companies and researchers
        in order to foster their role in the ecosystem or to
        signal specific projects/ventures."
        }
        ]
        ```
        }
```

*Policy target groups*
```
        Consider yourself an expert policy analyst tasked with
        reading documents related to Science Technology and
        Innovation (STI) policy.  Your goal is to comprehend
        and identify which of the below target group(s)
        the [REPLACE] initiative is relevant to and then
        assign them to relevant target groups based on the
        information within the given target group labels.
        These categories are provided to you in a JSON file
        with keys such as "target group code" for the ID of
        that target group, and "target group name" for the
        descriptive label of the target group.  Based on
        the text given to you, identify which of the given
        target groups types fit and are mentioned in the text,
        and identify the target group code with your short
        justification in English.  A text can be relevant
        to one or more target group, return only relevant
        matches.  You should structure your response as a
        JSON array where each object contains "TargetGroupID"
        as the key for the target group ID and "reason" as
        the key for your reasoning.  If you don't find any
        relevant target group, just say "n.a."
        Example response format:

        [
          {
            "TargetGroupID": "TG20",
            "reason": "reasoning..."
          },
          {
            "TargetGroupID": "TG9",
            "reason": "reasoning..."
          }
        ]

        If no relevant target group is found, the response
        should be:
        {
        "n.a.":  "No relevant target group found."
        }
        Here are the target group codes and names in a
        structured JSON file:
        ```json[
        {
```

```
      "target group code":   "TG20", "target group name":
      "Research and education organisations|Higher education
      institutes"
      },
      {
      "target group code":   "TG21", "target group name":
      "Research and education organisations|Public research
      institutes"
      },
      {
      "target group code":   "TG22", "target group name":
      "Research and education organisations|Private research
      and development lab" },
      {
      "target group code":   "TG9", "target group name":
      "Researchers, students and teachers|Established
      researchers"
      },
      {
      "target group code":   "TG11", "target group name":
      "Researchers, students and teachers|Postdocs and other
      early-career researchers"
      },
      {
      "target group code":   "TG41", "target group name":
      "Researchers, students and teachers|Programme managers
      and other research support staff"
      },
      {
      "target group code":   "TG10", "target group name":
      "Researchers, students and teachers|Undergraduate and
      master students"
      },
      {
      "target group code":   "TG38", "target group name":
      "Researchers, students and teachers|Secondary
      education students"
      },
      {
      "target group code":   "TG12", "target group name":
      "Researchers, students and teachers|PhD students"
      },
      {
      "target group code":   "TG13", "target group name":
      "Researchers, students and teachers|Teachers"
      },
      {
      "target group code":   "TG29", "target group name":
      "Firms by size|Firms of any size"
      },
      {
      "target group code":   "TG30", "target group name":
      "Firms by size|Micro-enterprises"
      },
      {
      "target group code":   "TG31", "target group name":
      "Firms by size|SMEs"
      },
      {
```

```
        "target group code":  "TG32", "target group name":
        "Firms by size|Large firms"
        },
        {
        "target group code":  "TG33", "target group name":
        "Firms by size|Multinational enterprises"
        },
        {
        "target group code":  "TG25", "target group name":
        "Firms by age|Firms of any age"
        },
        {
        "target group code":  "TG26", "target group name":
        "Firms by age|Nascent firms (0 to less than 1 year
        old)"
        },
        {
        "target group code":  "TG27", "target group name":
        "Firms by age|Young firms (1 to 5 years old)"
        },
        {
        "target group code":  "TG28", "target group name":
        "Firms by age|Established firms (more than 5 years
        old)"
        },
        {
        "target group code":  "TG34", "target group name":
        "Intermediaries|Incubators, accelerators, science
        parks or technoparks" },
        {
        "target group code":  "TG35", "target group name":
        "Intermediaries|Technology transfer offices"
        },
        {
        "target group code":  "TG36", "target group name":
        "Intermediaries|Industry associations"
        },
        {
        "target group code":  "TG37", "target group name":
        "Intermediaries|Academic societies / academies"
        },
        {
        "target group code":  "TG42", "target group name":
        "Intermediaries|Non-governmental organisations (NGOs)"
        },
        {
        "target group code":  "TG40", "target group name":
        "Governmental entities|International entity"
        },
        {
        "target group code":  "TG23", "target group name":
        "Governmental entities|National government"
        },
        {
        "target group code":  "TG24", "target group name":
        "Governmental entities|Subnational government"
        },
        {
        "target group code":  "TG18", "target group name":
```

```
1242        "Economic actors (individuals)|Entrepreneurs"
1243        },
1244        {
1245        "target group code":  "TG17", "target group name":
1246        "Economic actors (individuals)|Private investors"
1247        },
1248        {
1249        "target group code":  "TG19", "target group name":
1250        "Economic actors (individuals)|Labour force in
1251        general"
1252        },
1253        {
1254        "target group code":  "TG14", "target group name":
1255        "Social groups especially emphasised|Women"
1256        },
1257        {
1258        "target group code":  "TG15", "target group name":
1259        "Social groups especially emphasised|Disadvantaged and
1260        excluded groups" },
1261        {
1262        "target group code":  "TG16", "target group name":
1263        "Social groups especially emphasised|Civil society"
1264        }
1265        ]
1266        ```
```

*Policy themes*
Consider yourself an expert policy analyst tasked with
reading documents related to Science, Technology, and
Innovation (STI) policy.  Your goal is to comprehend
and identify which of the below policy themes the
[REPLACE] initiative is relevant to and assign the
appropriate policy themes based on the information
within the given policy theme labels and policy theme
relevancy guiding questions.  These categories are
provided to you in a JSON file where you can find
the guiding "question" to ask before assigning the
policy theme.  Each policy theme includes a "label"
and a "code".  Based on these guidelines, identify
which policy theme "code" fits the given policy
theme name and related question, and provide a short
justification for your selection in English.  A
text can be relevant to one or more policy theme,
return only relevant matches.  Structure your
response as a JSON array where each object contains
"PolicyThemeCode" as the key for the policy theme and
"reason" as the key for your reasoning.  If you don't
find any relevant policy theme, just say "n.a."
Example response format:

```
[
  {
    "PolicyThemeCode": "TH26",
    "reason": "reasoning..."
  },
  {
    "PolicyThemeCode": "TH30",
    "reason": "reasoning..."
  }
]
```

```json
If no relevant policy theme is found, the response
should be:
{
"n.a.":  "No relevant policy theme found."
}
Here are the policy theme codes, labels, and their
defining questions in a structured JSON file:

```json [
{
"code":  "TH11",
"label":  "Governance|Governance debates",
"question":  "Briefly, what are the main ongoing
issues of debate around how STI policy is governed?"
},
{
"code":  "TH13",
"label":  "Governance|STI plan or strategy",
"question":  "What strategies or plans exist, if any,
to provide an overarching strategic direction to STI
policy?"
},
{
"code":  "TH9",
"label":  "Governance|Horizontal policy coordination",
"question":  "What arrangements exist to support
cross-government coordination in STI policy?"
},
{
"code":  "TH14",
"label":  "Governance|Strategic policy intelligence",
"question":  "What arrangements or policy initiatives
exist to strengthen the evidence base for STI
policy-making and governance (besides evaluation and
impact assessment)?"
},
{
"code":  "TH15",
"label":  "Governance|Evaluation and impact
assessment",
"question":  "What arrangements exist to initiate,
reform, perform or encourage the use of STI evaluation
and impact assessment?"
},
{
"code":  "TH63",
"label":  "Governance|International STI governance
policy",
"question":  "What arrangements exist to support the
international governance of STI policy (e.g.  joint
strategies and agreements, horizontal coordination or
regulatory oversight bodies)?"
},
{
"code":  "TH16",
"label":  "Public research system|Public research
debates",
"question":  "Briefly, what are the main ongoing
policy debates around government support for the
```

```
        public research system?"
        },
        {
        "code":  "TH18",
        "label":  "Public research system|Public research
        strategies",
        "question":  "What strategies, roadmaps or plans
        exist, if any, to provide strategic direction to
        research policy?"
        },
        {
        "code":  "TH19",
        "label":  "Public research system|Competitive research
        funding",
        "question":  "What are the main competitive schemes
        and programmes for funding research in universities
        and public research institutes?"
        },
        {
        "code":  "TH20",
        "label":  "Public research system|Non-competitive
        research funding",
        "question":  "What are the main non-competitive
        schemes and programmes for funding research in
        universities and public research institutes?"
        },
        {
        "code":  "TH27",
        "label":  "Public research system|Third-party
        funding",
        "question":  "What policy initiatives exist to
        promote funding of public research from non-government
        sources?"
        },
        {
        "code":  "TH22",
        "label":  "Public research system|Structural change in
        the public research system",
        "question":  "What policy initiatives exist, if any,
        to support or lead structural changes in the public
        research system?"
        },
        {
        "code":  "TH106",
        "label":  "Public research system|Digital
        transformation of research-performing organisations",
        "question":  "What policy initiatives exist, if
        any, to help research-performing organisations
        upgrade their use of digital technologies (e.g.
        high-performance computing, big data analytics and
        artificial intelligence)?"
        },
        {
        "code":  "TH107",
        "label":  "Public research system|Open and enhanced
        access to publications",
        "question":  "What policy initiatives exist to support
        open and enhanced access to publications?"
        },
```

```
{
"code":  "TH108",
"label":  "Public research system|Open and enhanced
access to research data",
"question":  "What policy initiatives exist to support
open access to research data?"
},
{
"code":  "TH24",
"label":  "Public research system|Research and
technology infrastructures",
"question":  "What are the main policy initiatives for
funding the construction, operation of, and access to
research and technology infrastructures?"
},
{
"code":  "TH25",
"label":  "Public research system|Internationalisation
in public research",
"question":  "What are the main policy initiatives for
promoting internationalisation in public research?"
},
{
"code":  "TH26",
"label":  "Public research system|Cross-disciplinary
research",
"question":  "What are the main policy initiatives
for promoting inter, multi and transdisciplinary
research?"
},
{
"code":  "TH23",
"label":  "Public research system|High-risk
high-reward research",
"question":  "What policy initiatives exist, if any,
offering dedicated support to high-risk high-reward
research?"
},
{
"code":  "TH21",
"label":  "Public research system|Research integrity
and reproducibility",
"question":  "What are the main policy initiatives for
promoting research integrity and reproducibility?"
},
{
"code":  "TH109",
"label":  "Public research system|Research security",
"question":  "What are the main policy initiatives for
promoting research security and academic freedom?"
},
{
"code":  "TH28",
"label":  "Innovation in firms and innovative
entrepreneurship|Business innovation policy debates",
"question":  "Briefly, what are the main ongoing
policy debates around government support to business
innovation and innovative entrepreneurship?"
},
```

```
{
"code":  "TH30",
"label":  "Innovation in firms and innovative
entrepreneurship|Business innovation policy
strategies",
"question":  "What strategies or plans exist, if any,
to strategically direct government support to business
innovation and/or innovative entrepreneurship?"
},
{
"code":  "TH31",
"label":  "Innovation in firms and innovative
entrepreneurship|Financial support to business R&D
and innovation",
"question":  "What are the main policy initiatives
for providing financial support to business R&D and
innovation?"
},
{
"code":  "TH32",
"label":  "Innovation in firms and innovative
entrepreneurship|Non-financial support to business
R&D and innovation",
"question":  "What are the main policy initiatives for
providing non-financial support to business R&D and
innovation?"
},
{
"code":  "TH38",
"label":  "Innovation in firms and innovative
entrepreneurship|Access to finance for innovation",
"question":  "What policy initiatives exist to promote
firms' access to finance for innovation?"
},
{
"code":  "TH34",
"label":  "Innovation in firms and innovative
entrepreneurship|Entrepreneurship capabilities and
culture",
"question":  "What policy initiatives exist to foster
a spirit and culture of entrepreneurship in business
or in individuals and to provide them with appropriate
skills?"
},
{
"code":  "TH33",
"label":  "Innovation in firms and innovative
entrepreneurship|Stimulating demand for innovation
and market creation",
"question":  "What policy initiatives exist to
stimulate demand for firms' innovations and to support
market-creating innovation?"
},
{
"code":  "TH82",
"label":  "Innovation in firms and innovative
entrepreneurship|Digital transformation of firms",
"question":  "What policy initiatives exist, if
any, to help firms upgrade their organisational
```

```
and technological capabilities to undergo digital
transformation?"
},
{
"code":  "TH36",
"label":  "Innovation in firms and innovative
entrepreneurship|Foreign direct investment",
"question":  "What policy initiatives exist to attract
knowledge-intensive foreign direct investment and
promote transfers to domestic firms?"
},
{
"code":  "TH35",
"label":  "Innovation in firms and innovative
entrepreneurship|Targeted support to SMEs and young
innovative enterprises",
"question":  "What are the main policy initiatives
specifically targeting research and innovation
activities in SMEs, start-ups and young innovative
enterprises?"
},
{
"code":  "TH39",
"label":  "Knowledge exchange and
co-creation|Knowledge exchange and co-creation
debates",
"question":  "Briefly, what are the main ongoing
policy debates around policy for knowledge exchange
and co-creation involving academia, industry,
government and society?"
},
{
"code":  "TH41",
"label":  "Knowledge exchange and
co-creation|Knowledge exchange and co-creation
strategies",
"question":  "What strategies or

plans exist, if any, to strategically direct
government support for knowledge exchange and
co-creation?"
},
{
"code":  "TH42",
"label":  "Knowledge exchange and
co-creation|Collaborative research and innovation",
"question":  "What are the main policy initiatives to
promote collaboration between public researchers and
other stakeholders, including business and citizens?"
},
{
"code":  "TH47",
"label":  "Knowledge exchange and co-creation|Cluster
policies",
"question":  "What policy initiatives exist to promote
geographical and/or thematic innovative clusters?"
},
{
"code":  "TH43",
```

```
            "label":  "Knowledge exchange and
            co-creation|Commercialisation of public research
            results",
            "question":  "What policy initiatives exist to
            encourage commercialisation of public research
            results?"
            },
            {
            "code":  "TH44",
            "label":  "Knowledge exchange and
            co-creation|Inter-sectoral mobility",
            "question":  "What policy initiatives exist to
            encourage mobility of human resources between the
            public and private sectors?"
            },
            {
            "code":  "TH46",
            "label":  "Knowledge exchange and
            co-creation|Intellectual property rights in public
            research",
            "question":  "What policy initiatives exist to ensure
            intellectual property rights in public research are
            conducive to promoting innovation?"
            },
            {
            "code":  "TH48",
            "label":  "Human resources for research and
            innovation|STI human resources debates",
            "question":  "Briefly, what are the main ongoing
            policy debates around government support for human
            resources for research and innovation?"
            },
            {
            "code":  "TH50",
            "label":  "Human resources for research and
            innovation|STI human resources strategies",
            "question":  "What strategies or plans exist, if any,
            to strategically direct government support to human
            resources for research and innovation?"
            },
            {
            "code":  "TH51",
            "label":  "Human resources for research and
            innovation|STEM skills",
            "question":  "What are the main policy initiatives for
            nurturing general STEM skills?"
            },
            {
            "code":  "TH52",
            "label":  "Human resources for research and
            innovation|Doctoral and postdoctoral researchers",
            "question":  "What policy initiatives exist to
            specifically support doctoral and postdoctoral
            research and education?"
            },
            {
            "code":  "TH53",
            "label":  "Human resources for research and
            innovation|Research careers",
```

```
"question":  "What policy initiatives exist to make
research careers more attractive?"
},
{
"code":  "TH55",
"label":  "Human resources for research and
innovation|International mobility of human resources",
"question":  "What policy initiatives exist to
encourage international mobility of researchers?"
},
{
"code":  "TH54",
"label":  "Human resources for research and
innovation|Equity, diversity and inclusion (EDI)",
"question":  "What policy initiatives exist to promote
the participation of women and other under-represented
groups in research and innovation activities?"
},
{
"code":  "TH56",
"label":  "Research and innovation for society|Policy
debates on innovation for societal challenges",
"question":  "Briefly, what are the current main
policy debates around how policy for research and
innovation can help address societal challenges?  If
applicable, please elaborate on how the Sustainable
Development Goals (SDGs) are being incorporated into
STI policy objectives, design and implementation."
},
{
"code":  "TH58",
"label":  "Research and innovation for
society|Research and innovation for society strategy",
"question":  "What strategies or plans exist, if
any, to strategically direct government support for
research and innovation specifically targeted at
societal well-being and cohesion?"
},
{
"code":  "TH91",
"label":  "Research and innovation for
society|Mission-oriented innovation policies",
"question":  "What cross-government initiatives exist,
if any, to coordinate and jointly operate different
policy initiatives to achieve ambitious goals within a
defined timeframe and to address a societal challenge
(e.g.  the EU missions { Climate Change, Cancer,
Oceans, Cities, Soil)?"
},
{
"code":  "TH89",
"label":  "Research and innovation for society|Ethics
of emerging technologies",
"question":  "What policy initiatives exist, if any,
to address ethical challenges raised by emerging
technologies (e.g.  artificial intelligence,
biotechnology, quantum computing)?"
},
{
```

```
"code":  "TH61",
"label":  "Research and innovation for
society|Research and innovation for developing
countries",
"question":  "What policy initiatives exist, if any,
specifically dedicated to supporting research and
innovation in developing and less technologically
advanced countries?"
},
{
"code":  "TH65",
"label":  "Research and innovation for
society|Multi-stakeholder engagement",
"question":  "What policy initiatives exist to promote
a broad and diversified public engagement in research
and innovation activities and policy making?"
},
{
"code":  "TH66",
"label":  "Research and innovation for
society|Science, technology and innovation culture",
"question":  "What are the main policy initiatives for
building understandings and common STI culture across
technical communities and citizens?"
},
{
"code":  "TH101",
"label":  "Net zero transitions|Net zero transitions
policy debates",
"question":  "Briefly, what are the current main
policy debates around how net zero emission targets
are being incorporated into STI policy objectives,
design and implementation?"
},
{
"code":  "TH102",
"label":  "Net zero transitions|Government
capabilities for net zero transitions",
"question":  "What reforms, if any, have been
implemented to improve the operation and capabilities
of STI ministries and agencies to better address net
zero transitions?"
},
{
"code":  "TH92",
"label":  "Net zero transitions|Net zero transitions
in energy",
"question":  "What policy initiatives, if any, aim
specifically to support research and innovation
for net-zero carbon ambitions in the energy sector
(electricity and heat)?"
},
{
"code":  "TH103",
"label":  "Net zero transitions|Net zero transitions
in transport and mobility",
"question":  "What policy initiatives, if any, aim
specifically to support research and innovation
for net-zero carbon ambitions in the transport and
```

```
mobility sectors?"
},
{
"code":  "TH104",
"label":  "Net zero transitions|Net zero transitions
in food and agriculture",
"question":  "What policy initiatives, if any, aim
specifically to support research and innovation for
net-zero carbon ambitions in the food and agriculture
sectors?"
},
{
"code":  "TH105",
"label":  "Net zero transitions|STI policies for net
zero",
"question":  "Please link to this question policies in
other sections of the questionnaire (i.e.  outside of
this module) that prominently aim to achieve net zero
carbon ambitions."
}
]```
```

*Budget*
```
Using the information provided, your task is to
determine if there is any monetary information such
as budget or expenditure related to the [REPLACE]
initiative.  Make your response in English.  If there
is no information, respond with "No information" and
do not elaborate.  Answer in English and provide your
answers in a JSON structured format.
Example JSON response:
```json[
{
"monetaryInformation":  "Budget of $10 million
allocated for research and development."
},
{
"monetaryInformation":  "Budget of $5 million
allocated for implementation."
}
]
```
If no monetary information is found, the response
should be:
```json
{
"monetaryInformation":  "No information"
}
```
```

*Evaluation report*
```
Using the information provided, your task is to
determine if the [REPLACE] initiative has been
evaluated and if an evaluation report exists.  If
evaluation is not mentioned, respond with "No
information" and do not elaborate.  Structure
your information as a JSON file where the key is
"evaluation name" and the value is "the information of
the found evaluation" in English.  Avoid using jargon;
```

```
be concise and clear, delivering only information
retrieved from the text.
Example JSON response:
```json [
{
"evaluationName":  "Mid-Term Evaluation Report",
"information":  "The mid-term evaluation report
conducted in 2023 assesses the effectiveness and
impact of the policy initiative."
}
]
```

If no evaluation information is found, the response
should be:
```json
{
"evaluationName":  "n.a.",
"information":  "No information"
}
```
```

## B.4  SYSTEM PROMPT FOR FINE-TUNING CAUSAL MODELS

```
You are an AI assistant trained to classify text about
Science, Technology, and Innovation (STI) policy.
Your task is to identify the most relevant category
labels from the three dimensions below:
1.  Policy Instruments (PI)
2.  Policy Target Groups (TG)
3.  Policy Themes (TH)
Each category has a list of definitions provided.
Based on the input text, identify and return only the
**Labels** of items that are clearly relevant to the
content.  Your answer should be only a **flat list
of matching labels**.  DO NOT provide any additional
text.
Policy Instruments (PI) labels and definitions:
{ pi_definitions }
Policy Target Groups (TG) labels and definitions:
{ tg_definitions }
Policy Themes (TH) labels and definitions:
{ th_definitions }
```

