# OpenReview forum: "AI Respondents for Policy Monitoring: From Data Extraction to AI-Driven Survey Responses in the OECD STIP Compass"
_ICLR.cc/2026/Conference — Submitted to ICLR 2026_

### Official Review · Reviewer_PmP8 · 2025-10-23

**Soundness:** 2
**Presentation:** 2
**Contribution:** 2
**Rating:** 2
**Confidence:** 4

**Summary:**

The authors prompt LLMs to summarize and classify science, technology, and innovation policy documents, then analyze the differences between the LLM responses and human responses.

**Strengths:**

* There is substantial need to assess the utility of LLMs for analysis of policy documents

**Weaknesses:**

* The introduction does not explain the task clearly. What do these policy documents look like? What does "mapping" and "monitoring" these policies look like? That is, what are the inputs and outputs of the day-to-day tasks people in this field are trying to automate with LLMs?
* No agreement between the evaluator-LLMs and humans was calculated. Using an LLM as an evaluator is only okay if we have a measure of how well the evaluator LLM agrees with human evaluation judgments.
* No estimation of how hard humans find the task is included. The authors claim "achieving consistently high agreement is challenging and depends on interpretation, comprehension, and background knowledge" but do not demonstrate this by measuring expert agreement on policy instrument, target group, and theme labels.
* The classification agreement is calculated over "all overlapping cases where there is an overlap for at least one policy label" but should also be calculated over all cases.
* The cross-validation experiments are not defined in sufficient detail. How was the training data formatted? It couldn't have been the same as for the LLMs, given that RoBERTa has a tiny context window.
* The overlap analysis is too coarse grained. A finer grained analysis of what kinds of things LLMs and humans coincide on and what kinds of things they diverge on is needed. This can be seen in problematic claims like "LLMs tend to provide more detailed procedural and descriptive accounts, human experts emphasize the contextual and societal implications of policy initiatives" when the analyses show only that humans and LLMs diverge more on objectives than they do on description. A finer grained analysis that studies "procedural" and "societal" language in the summaries would be needed to support such claims.

**Questions:**

* Are all of the models in the cross-validation section fine-tuned? E.g., you fine-tuned GPT-4o 128k?

---

### Official Review · Reviewer_W83R · 2025-10-31

**Soundness:** 2
**Presentation:** 2
**Contribution:** 1
**Rating:** 0
**Confidence:** 4

**Summary:**

The paper focuses on assigning labels to Science, Technology, and Innovation (STI) policies for documents of OECD countries. It develops an LLM-based classification system that integrates intermediate validation steps -- like relevance and evidence scoring. They evaluate the system with an existing set of answers by human respondents.

**Strengths:**

The paper clearly has strengths. The overall topic is likely very relevant for a lot of people in the policy space. The use case is well-presented and worked out with a good amount of carefulness. It is fairly well-written.

**Weaknesses:**

However, this paper presents a clear mismatch with what I would consider relevant for an AI conference. It is an LLM engineering use case from which possibly policy-related researchers could be inspired in their respective publication outlets. It does not deliver a significant contribution (method, datasets, etc.) in the AI/ML space. It just applies an LLM to data. This shallowness in contribution is also reflected in the fact that only superficial literature is cited with no reference to an actual stream of literature in ML, where this would be relevant. I think any other evaluation is pointless, given that the paper belongs to a different community.

**Questions:**

What literature do you contribute to?
Which method or data innovation do you present?

---

### Official Review · Reviewer_jVSH · 2025-11-01

**Soundness:** 2
**Presentation:** 3
**Contribution:** 2
**Rating:** 4
**Confidence:** 3

**Summary:**

This paper presents a novel application of Large Language Models (LLMs) to automate data extraction and survey completion for the OECD STIP Compass, which monitors Science, Technology, and Innovation (STI) policies across countries. The authors develop a pipeline using GPT-4o with long-context in-context learning to extract structured information from policy documents and generate survey responses. Testing on six countries, they achieve 84-95% overlap with human respondents on structured policy indicators, though qualitative differences emerge in free-text fields where LLMs provide more procedural detail while humans emphasize contextual impacts.

**Strengths:**

The paper addresses a genuinely important problem in policy research: the labor-intensive nature of international policy monitoring. The methodology is reasonably transparent, including validation mechanisms and multiple evaluation approaches (overlap analysis, agreement scoring, cross-validation). The recognition that LLMs and human experts complement rather than substitute for each other is sophisticated and realistic. The finding that LLMs excel at structured extraction but differ from humans in interpretive framing aligns well with social science understanding of how context and tacit knowledge shape policy analysis. The study's acknowledgment of potential biases and "model collapse" risks demonstrates appropriate caution about AI-generated data quality.

**Weaknesses:**

The evaluation relies heavily on overlap metrics rather than assessing the quality or validity of policy interpretation. High agreement doesn't necessarily mean accurate understanding of policy intent or impact. the country selection is reasonable but also limits generalizability. The paper does not address LLMs possible systematically misunderstand of culturally-specific policy contextsThere is no discussion of how power dynamics and epistemological questions (who defines what counts as valid policy knowledge?) intersect with AI automation.

Also, policies in countries with less digital transparency, or older initiatives, will be systematically underrepresented. The maper measures web presence as much as policy content. This is a fundamental validity threat the paper barely acknowledges.

The paper lacks an assessment of whether LLM-generated policy classifications maintains the correlation structures, distributional properties, or causal relationships that policy researchers need. Overlap does not equate to statistical exchangeability.

The paper lacks a formal framework for weighting human vs. AI inputs. It has no analysis of when to trust which source, no error propagation model.

**Questions:**

How do you quantify and communicate the uncertainty in your LLM-generated policy labels? The binary validation (0/1 for "evidenced" and "relevant") seems crud. Could one use confidence scores or calibrated probabilities? How should policy analysts interpret disagreements between human and AI responses?Which types of policies or countries does the model systematically fail on? Can you provide instance-level confidence measures? You claim this enables 'scalable' policy monitoring, but you've only tested six countries and relied on expensive proprietary models (€447 for 845 samples). What's the actual cost-benefit at scale, and what happens when GPT-4o becomes unavailable or changes?

---

### Official Review · Reviewer_mvN5 · 2025-11-12

**Soundness:** 3
**Presentation:** 3
**Contribution:** 1
**Rating:** 2
**Confidence:** 3

**Summary:**

The paper introduces an AI-assisted data extraction pipeline for the OECD STIP Compass survey, which collects cross-country data on Science, Technology, and Innovation (STI) policies. The authors propose using Large Language Models (LLMs) as “AI respondents” that can automatically pre-fill structured survey fields (policy instruments, target groups, themes) from policy webpages using long-context in-context learning (ICL). A second LLM layer validates these outputs by assessing relevance and evidence.

The pipeline is tested across six OECD countries (Canada, Finland, Germany, South Korea, Spain and Turkey). Results show high agreement between AI- and human-filled structured codes (84–95%) and reasonable alignment in free-text descriptions (~74% high overlap). Textual differences reveal complementary strengths: LLMs generate procedural detail, while humans emphasize contextual and societal framing. Cross-validation (GPT-4o vs others) yields ~98 F1 micro for AI labels. The authors conclude that hybrid human: & AI survey workflows could reduce reporting burden and increase consistency, provided oversight avoids bias or model collapse from synthetic feedback loops.

**Strengths:**

**MAJOR STRENGHTS**

- **Useful + impactful for policy work**: a practical pipeline that can materially reduce survey burden and increase consistency in STI policy monitoring.
- **Well-written + easy to follow**: clear structure and framing; good figures/tables; reads smoothly even for non-specialists.
- **Interesting application**: treats LLMs as AI respondents to pre-fill structured policy surveys (not just generic classification).
- **Sound two-stage design**: long-context ICL for extraction + a validator LLM that scores relevance/evidence (mitigates hallucination/misreadings).
- **Solid empirical grounding**: multi-country evaluation with quantitative overlap/F1 results against human entries.
- **Human+AI complementarity surfaced**: LLMs add procedural/detail; humans add context/societal framing, which is a potentially useful insight that hints at a deeper human vs LLMs follow-up paper?
- **Generalizable blueprint**: pipeline looks readily adaptable to other policy domains (beyond OECD/STI) and to human–AI co-workflows.
- **Risk-aware**: discusses pitfalls (hallucination, synthetic-feedback/model-collapse), not just benefits.
- **Operationally realistic**: choices (long-context prompting, schema outputs, validator checks) feel deployable by policy teams.

**Weaknesses:**

**MAJOR WEAKNESS/MISMATCH**
- **Overall fit (why I list this as a poor contribution for ICLR):** While the paper is genuinely useful and seems to be well-executed for policy analysis, it reads primarily as an LLM application/pipeline for policy data collection (human+AI survey workflows) rather than as a core methodological contribution in ML. That makes it feel better suited to policy-oriented venues (AI for governance, digital government, measurement) than to ICLR’s typical focus on new learning algorithms, theory, or broadly generalizable methodology? I’m open to being convinced otherwise by the editor/chair, but my expertise and reviewing lens skew toward method innovation, not domain-specific deployment studies; I can’t fully judge the policy-side contribution. This mismatch, plus how different this paper is from the rest of the papers I'm currently reviewing for ICLR, drives my evaluation of contribution/venue fit.

**MINOR POINTS**
- **Make “causal models” in the K-fold section explicit**: the paper mentions using causal models for cross-validation, but please specify which models, what “causal” means here (e.g., structure priors? debiasing? just a model name?), and how they affect label quality or agreement.
- **Code and assets availability**: if a repo or artifacts (prompts, validators, evaluation scripts, country lists/URLs) are not publicly available, please release them; if they are, surface the link prominently (abstract or intro).
- **Broaden evaluation beyond overlap**: is it possible to add expert adjudication on a stratified sample (policy analysts rating correctness/justification), and report hallucination/error categories (fabricated initiatives, wrong mappings, stale links)? That would strengthen conclusions.
- **Human+AI workflow clarity**: it would be nice if there were a pipeline figure showing hand-offs (AI pre-fill → validator → human editor), with failure modes and fallback steps (what triggers manual review); just a suggestion.

**Questions:**

See points above.

---

### Meta-Review · Area_Chair_gUW8 · 2026-01-08

**Summary:**

The paper presents an LLM-based pipeline for automating structured information extraction and survey completion in the OECD STIP Compass, a large-scale international initiative for monitoring Science, Technology, and Innovation (STI) policies. The core idea is to treat Large Language Models as “AI respondents” that pre-fill structured survey fields (policy instruments, target groups, themes) from long policy documents using long-context in-context learning, followed by a secondary LLM that performs relevance and evidence validation. The system is evaluated on six OECD countries, reporting high overlap (84–95%) between AI-generated and human-generated structured indicators, along with qualitative differences in free-text descriptions. The authors argue that this hybrid human–AI workflow can improve efficiency and consistency in policy monitoring.

**Reviewer Concerns:**

The reviewers raised several substantial concerns that were not resolved to a level sufficient for acceptance.

1. A central and recurring concern is that the paper does not make a clear methodological, theoretical, or data contribution to machine learning research. The work is primarily an application and engineering pipeline that deploys existing LLM capabilities to a domain-specific policy monitoring task. Reviewers consistently noted that the paper reads as a policy or digital governance deployment study rather than a contribution aligned with ICLR’s focus on novel learning methods, theory, or broadly generalizable ML insights.

2. The empirical evaluation relies heavily on overlap and agreement scores between LLM-generated and human-generated labels. Reviewers emphasized that high overlap does not establish correctness, validity, or substantive understanding of policy content. There is no measurement of human inter-annotator agreement, no calibration of evaluator LLM judgments against human evaluations, and no expert adjudication of correctness. As a result, the evaluation does not convincingly demonstrate that the system produces reliable or policy-valid outputs.

3. The validation framework reduces relevance and evidence to binary signals, providing no principled way to quantify uncertainty, propagate errors, or decide when human judgment should override AI outputs. There is no formal framework for weighing human versus AI inputs, nor an analysis of systematic failure modes across policy types or countries.

4. Reviewers raised concerns that the method may primarily capture web presence rather than policy substance, systematically disadvantaging countries or initiatives with lower digital transparency. The paper does not assess whether AI-generated labels preserve distributional properties, correlation structures, or causal relationships required for downstream policy research. Cultural and contextual misunderstandings by LLMs are also insufficiently addressed.

**Reviewer Scores:**

The majority of reviewers rated the contribution as poor to fair, with multiple explicit reject recommendations, including one strong reject. While soundness and presentation were generally assessed as fair to good, concerns about contribution, evaluation rigor, and venue fit dominated the final assessments.

---

### Decision · Program_Chairs · 2026-01-26

Reject